# Effects of Mink-Origin *Enterococcus faecium* on Growth Performance, Antioxidant Capacity, Immunity, and Intestinal Microbiota of Growing Male Minks

**DOI:** 10.3390/ani14142120

**Published:** 2024-07-20

**Authors:** Lin Cao, Fengxue Sun, Qifeng Ren, Ziyi Jiang, Jian Chen, Yalin Li, Lihua Wang

**Affiliations:** College of Animal Science and Technology, Qingdao Agricultural University, Qingdao 266109, China; 13963906513@163.com (L.C.); sunfengxue2022@163.com (F.S.); ren122633@163.com (Q.R.); jiangziyi2024@163.com (Z.J.); 17660987851@163.com (J.C.); liyalin827@163.com (Y.L.)

**Keywords:** mink, *Enterococcus faecium*, growth performance, antioxidant capacity, immunity, intestinal flora

## Abstract

**Simple Summary:**

*Enterococcus faecium*, a lactic acid bacterium, is approved for use as a direct-fed microbial by the Ministry of Agriculture (MOA) in China. However, some research reports indicate that certain strains of *Enterococcus faecium* have developed antibiotic resistance. Moreover, some strains of *Enterococcus faecium* carry virulence genes and are considered to be conditional pathogens. Therefore, selecting safe and reliable strains of *Enterococcus faecium* as probiotics for livestock and poultry is particularly important. This experiment used *Enterococcus faecium* isolated from the intestinal contents of healthy minks to evaluate the effects of *Enterococcus faecium* on the growth performance, antioxidant capacity, immunity, and gut microbiota composition of growing male minks.

**Abstract:**

The purpose of this experiment was to explore the effects of dietary *Enterococcus faecium* (EF) on the growth performance, antioxidant capacity, immunity, and intestinal microbiota of growing male minks. A total of 60 male Regal White minks at 12 weeks of age were randomly assigned to two groups, each with 15 replicates of two minks per replicate. The minks in two groups were fed the basal diets and the basal diets with viable *Enterococcus faecium* (more than 10^7^ cfu/kg of diet), respectively. Compared with the minks in control, *Enterococcus faecium* minks had heavier body weight (BW) at week 4 and week 8 of the study (*p* < 0.05), greater average daily gain (ADG), and a lower feed/gain ratio (F/G) of male minks during the initial 4 weeks and the entire 8-week study period (*p* < 0.05). Furthermore, *Enterococcus faecium* increased the apparent digestibility of crude protein (CP) and dry matter (DM) compared to the control (*p* < 0.05). Moreover, *Enterococcus faecium* enhanced the serum superoxide dismutase (SOD) activity and decreased the malondialdehyde (MDA) contents (*p* < 0.05). The results also confirmed that *Enterococcus faecium* increased the levels of serum immunoglobulin A (IgA), immunoglobulin G (IgG), and the concentrations of secretory immunoglobulin A (SIgA) in the jejunal mucosa while decreasing the interleukin-8 (IL-8) and interleukin-1β (IL-1β) levels in the jejunal mucosa (*p* < 0.05). Intestinal microbiota analysis revealed that *Enterococcus faecium* increased the species numbers at the OUT level. Compared with the control, *Enterococcus faecium* had significant effects on the relative abundance of *Paraclostridium*, *Brevinema*, and *Comamonas* (*p* < 0.05). The results showed that *Enterococcus faecium* could improve the growth performance, increase the antioxidant capacity, improve the immunity of growing male minks, and also modulate the gut microbiota.

## 1. Introduction

Probiotics, as defined by the FAO/WHO, are live microorganisms that confer health benefits to the host when administered in adequate amounts [1,2]. Lactic acid bacteria are widely utilized as probiotics across various applications. These heterotrophic anaerobes ferment carbohydrates into lactic acid, an antibacterial compound that inhibits the growth of acid-sensitive harmful bacteria by lowering the pH value [3]. In addition to lactic acid, lactic acid bacteria also produce bacteriocins, mannitol, fatty acids, and exopolysaccharides, all of which are associated with beneficial physiologic effects [4]. Recognized as safe for consumption (GRAS), lactic acid bacteria have been extensively used as probiotics in the livestock and poultry industries [5,6,7].

*Enterococcus faecium*, a type of lactic acid bacterium, is naturally present in the digestive tracts of animals, soil, and water. It is resistant to heat, gastric acids, and bile salts [8]. In China, it was approved for use as a direct-fed microorganism by the Ministry of Agriculture (MOA) in 2003. Previous studies on pigs and chickens have verified that *Enterococcus faecium* can improve growth performance, enhance immunity, and regulate intestinal flora [9,10].

The employment of *Enterococcus faecium* as a probiotic is still controversial. *Enterococcus faecium* exhibits essential probiotic properties, including heat resistance, acid tolerance, bile salt resistance, adhesion capabilities, and the ability to inhibit bacterial growth [11,12]. However, its strong adaptability increases the potential for *Enterococcus faecium* to develop antibiotic resistance. Particularly, overuse of antibiotics has caused a serious problem of antibiotic resistance in some strains of *Enterococcus faecium* [13,14]. Furthermore, the resistance is encoded on plasmids, which makes it susceptible to the potential for spread through horizontal gene transfer [15]. Amaral et al. [16] suggested that *Enterococcus faecium* strains isolated from food, commensal organisms, and the environment carry a low risk. As a normal component of the intestinal flora in animals, *Enterococcus faecium* should not be entirely ruled out as a probiotic candidate due to the pathogenicity of a few strains [17]. Therefore, identifying reliable sources for the isolation of potential probiotic strains is crucial. Considering that probiotic properties are related to host specificity, strains isolated from the host species may exhibit more advantageous probiotic effects compared to those derived from other origins [18,19]. 

This experiment utilized the *Enterococcus faecium* strain isolated from the intestinal contents of healthy minks to explore its effects on the growth performance, antioxidant capacity, immunity, and gut microbiota composition of growing male minks.

## 2. Material and Methods

### 2.1. Ethics

The Animal Care and Use Committee of Animal Science and Technology at Qingdao Agricultural University reviewed and approved the experimental protocol (approval number DKY20230524-1, date: 24 May 2023).

### 2.2. Enterococcus faecium

*Enterococcus faecium* (EF) was previously isolated from the rectal contents of minks, identified through 16S rRNA gene sequence analysis, and is preserved in the China General Microbiological Culture Collection Center (Accession No. 29262). The isolated strain of *Enterococcus faecium* was inoculated into the De Man, Rogosa, and Sharpe (MRS) substrate and cultivated at 37 °C for 24 h. The viable *Enterococcus faecium* in the suspension was not less than 10^9^ cfu/mL. The prepared bacterial suspension was stored at 4 °C for use in the present study, prepared once every 20 days to ensure that the viable count of *Enterococcus faecium* remained above 10^7^ cfu/mL.

### 2.3. Experimental Design and Animal Management

The research was performed at a commercial mink farm located in Haiyang, Yantai, Shandong Province. A total of 60 male Regal White minks at the age of 12 weeks with similar body weight (BW) were randomly assigned to two groups, each with 15 replicates of two minks. The minks in two groups were fed the basal diets and the basal diets with viable *Enterococcus faecium* probiotics (containing viable *Enterococcus faecium* at more than 10^7^ cfu/kg of diet), respectively. The paste diets were composed of sea fish and byproducts, chicken byproducts, and egg products. The diets were formulated on the farm based on commercial recommendations. The detailed composition of the experimental diets and the nutrient levels are presented in Table 1. Following a one-week adaptation period, the experiment was conducted for a duration of eight weeks.

All minks were housed in metallic cages arranged within an open-sided double-row shed. Each cage was made of identical wire mesh, measured 30 × 75 × 45 cm (width × length × height), and housed a pair of minks. The minks were fed twice daily at 6:30 a.m. and 3:30 p.m., respectively, throughout the study period. Each cage was equipped with a water drinker, ensuring minks enjoyed unrestricted access to drinking water. The recorded ambient temperature was 26.24 ± 3.31 °C, and the relative humidity was 65.27 ± 3.54% throughout the study. The light schedule adhered to the natural light regime.

### 2.4. Sample and Data Collection

#### 2.4.1. Growth Performance

All minks were individually weighed at the beginning (week 0), week 4, and week 8 of the study to determine the initial (week 0), week 4, and final (week 8) body weight. In addition, the feed provided and the leftovers were accurately weighed and recorded over three consecutive days each week throughout the experimental period. The average daily gain (ADG), average daily feed intake (ADFI), and feed-to-gain ratio (F/G) for the minks were calculated.

#### 2.4.2. Apparent Digestibility of Nutrients

A digestive experiment was conducted at week 7 of the study, using the endogenous indicator method to determine the apparent digestibility of nutrients. The digestive experiment lasted for 3 days and involved the collection of both diet and fecal samples. In each group, six fecal samples were collected, with each 3-day cumulative fecal sample being mixed and weighing approximately 200 g. Concurrently, diets for each group were sampled daily over the 3 days prior to feeding the minks, and these samples were then pooled to obtain representative diet samples. All diet and fecal samples were dried to a constant weight in a draft oven at 65 °C to obtain air-dried samples and determine the initial moisture content. The air-dried samples were ground in a 1 mm screen mill. The dry matter (DM), crude ash, hydrochloric acid insoluble ash, crude protein (CP), and ether extract (EE) in these diet and fecal samples were analyzed according to GB/T 6435-2014, GB/T 6438-2007, GB/T 23742-2009, GB/T 6432-2018, and GB/T 6433-2006, respectively.
Nutrient apparent digestibility (%) = 100% − (A1/A2) × (B2/B1)

A1: the content of hydrochloric acid-insoluble ash in the diet; A2: the content of hydrochloric acid-insoluble ash in the feces samples; B1: the content of a certain nutrient in the diet; B2: the content of a certain nutrient in the feces sample.

#### 2.4.3. Immunity and Antioxidant Capacity

At the end of the study, eight minks per group were randomly selected. Blood was collected via heart puncture, and the minks were subsequently euthanized to obtain 2–5 g of jejunum mucosal tissue. Serum was obtained from blood samples by centrifugation at 3000× *g* at 4 °C for 10 min. Serum levels of immunoglobulins (IgA, IgG, and IgM), total antioxidant capacity (T-AOC), glutathione peroxidase (GSH-pX), superoxide dismutase (SOD), and malondialdehyde (MDA) were measured. Jejunum mucosal tissue samples were diluted with 0.9% saline solution (1:9 weight/volume ratio) and then homogenized using a grinder. The homogenates were centrifuged at 3500× *g* for 10 min at 4 °C to obtain the supernatant, which was then used to analyze the contents of secretory immunoglobulin A (SIgA) and cytokines (IL-1β, IL-8, IL-10, IL-2, IL-6, IL-12, TNF-α, and IFN-γ). All these immune indicators were measured using ELISA kits manufactured by the Jiancheng Biological Engineering Research Institute in Nanjing, China. T-AOC and the activity of GSH-pX were determined using the colorimetric method. The activity of T-SOD was measured by the hydroxylamine method, and the MDA was determined using the thiobarbituric acid (TBA) method. All the antioxidant indicator assay kits used were manufactured by the Jiancheng Biological Engineering Research Institute in Nanjing, China.

#### 2.4.4. Intestinal Microbiota

Concurrently with the collection of jejunal mucosal tissues, rectal mucosal swabs were obtained. DNA from rectal mucosal swabs was extracted to determine intestinal flora, employing the method delineated by Li et al. [20]. The extracted genomic DNA was detected, amplified (ABI GeneAmp&reg, 9700), purified, recovered (Axyprep DNA Gel Extraction Kit, Axygen, Union City, CA, USA), quantified (QuantiFluor™-ST Blue Fluorescence Quantification System, Promega, Madison, WI, USA), and sequenced (Illumina MiSeq pE 300 platform) [21].

### 2.5. Statistical Analysis

The data on growth performance, immunity, and antioxidant capacity were analyzed by a Student’s *t*-test using SPSS 25.0 (SPSS Institute, Inc., Chicago, IL, USA), and the results were presented as means ± standard deviation. *p* < 0.05 means a significant difference.

The data of the intestinal microbiota were analyzed on the I-Sanger cloud platform. FLASH 1.2.11 software performs paired-end sequencing. The differences in alpha diversity indices and species composition were statistically analyzed through the Student’s *t*-test. The Spearman correlation coefficient was used to analyze the correlation between the intestinal flora and the immune indicators of the jejunal mucosa in growing male minks.

## 3. Results

### 3.1. Growth Performance

The *Enterococcus faecium* had significant effects on body weight, ADG, and F/G (*p* < 0.05, Table 2), but the effects were not observed on ADFI (*p* > 0.05). Compared with the control minks, EF minks had higher BW (*p* < 0.05) at week 4 and week 8 of the study, greater ADG (*p* < 0.05), and less F/G (*p* < 0.05) during the first 4 weeks and the entire 8 weeks of the study.

### 3.2. Apparent Digestibility of Nutrients

*Enterococcus faecium* had effects on the apparent digestibility of DM and CP (*p* < 0.05, Table 3), but the effects were not observed on the apparent digestibility of EE and Ash (*p* > 0.05). Compared with the control, EF increased the apparent digestibility of DM and CP (*p* < 0.05).

### 3.3. Serum Antioxidant Indexes

The *Enterococcus faecium* had effects on serum SOD activity and MDA level (*p* < 0.05, Table 4), but the effects were not observed on serum T-AOC level and GSH-pX activity (*p* > 0.05). Compared with the control, EF increased SOD activity (*p* < 0.05) and decreased MDA levels (*p* < 0.05). 

### 3.4. Serum Immune Indexes 

*Enterococcus faecium* had effects on serum IgA and IgG levels (*p* < 0.05, Table 5), but the effects were not observed on serum IgM (*p* > 0.05). Compared with the control, EF increased serum IgA and IgG levels (*p* < 0.05).

### 3.5. Jejunum Mucosal Immune Indexes

The *Enterococcus faecium* had significant effects on the levels of IL-8, IL-1β, and SIgA in the jejunum mucosal (*p* < 0.05, Table 6), but the effects were not observed on other indexes (*p* > 0.05). Compared with the control, EF significantly increased SIgA levels (*p* < 0.05) while decreasing IL-8 and IL-1β levels (*p* < 0.05).

### 3.6. Intestinal Microbiota

As shown in Figure 1, the sequencing saturation curve tends to flatten out at the end, indicating that all samples have sufficient sequencing depth. There were no differences among the two groups in the ACE, Chao, Shannon, and Sobs indices (*p* > 0.05, Figure 2A). However, as depicted in Figure 2B, the control group contained 718 species at the OUT level, while the EF group had 898 species at the OUT level, and 405 species were found to co-occur across both groups.

At the phylum level, *Proteobacteria*, unclassified_k__norank_d__Bacteria, *Firmicutes*, *Cyanobacteria*, and *Bacteroidota* were the top five dominant phyla in the control group (Figure 3A). *Firmicutes*, *Proteobacteria*, unclassified_k__norank_d__*Bacteria*, *Bacteroidota*, and *Actinobacteriota* were the top five dominant phyla in the EF group. At the genus level, unclassified_k__norank_d__Bacteria, *Mycoplasma*, *Sphingobium*, *Acinetobacter*, and *Enterobacter* were the top five dominant genera in the control group (Figure 3B). *Mycoplasma*, *Lactococcus*, *Sphingobium*, *Acinetobacter*, and unclassified_k__norank_d__Bacteria were the top five dominant genera in the EF group. Compared with the control, EF increased the relative abundance of *Paraclostridium*, *Brevinema*, and *Comamonas* (*p* < 0.05, Figure 3C).

### 3.7. Correlation Analysis

A Spearman correlation analysis was conducted to assess the relationship between the top 10 bacterial genera and the intestinal immune status of developing male minks (Figure 4). *Escherichia-Shigella* was positively correlated with IL-6. *Mycoplasma* was positively correlated with IL-2. *Acinetobacter* was positively correlated with TNF-α and negatively correlated with INF-γ. *Sphingobium* and *Sphingomonas* were negatively correlated with IL-6.

## 4. Discussion

One of the major findings of the current study was that *Enterococcus faecium* improved ADG in mink aged 12–16 weeks. It appears that the improvement in ADG was only evident during the initial 4 weeks of the study. Between week 5 and week 8, there was no difference in ADG between the control minks and those fed with *Enterococcus faecium*, indicating the short-term effects of *Enterococcus faecium* on growth performance. However, this short-term effect on ADG resulted in minks fed with *Enterococcus faecium* having a greater BW at both week 4 and week 8 of the study. Additionally, the increased ADG was associated with an improvement in the feed/gain ratio (F/G), but there was no associated change in average daily feed intake (ADFI). This suggests that minks fed *Enterococcus faecium* exhibited efficient dietary utilization. The results were consistent with those of previous studies on pigs [22,23], which found that supplementation with *Enterococcus faecium* increased growth performance and feed efficiency. Similarly, a study on rabbits [24] showed that supplementation with *Enterococcus faecium* could enhance the live weight and ADG. In the current study, *Enterococcus faecium* increased the apparent digestibility of DM and CP. The results are consistent with previous research on pigs, which has shown that *Enterococcus faecium* enhances nutrient digestibility [25,26]. It suggests that the improvement in the growth performance of minks is attributed to *Enterococcus faecium* enhancing the apparent digestibility of nutrients. Chen et al. [27] have demonstrated that *Enterococcus faecium* has a positive influence on the digestibility of DM in pigs. As a lactic acid bacterium, *Enterococcus faecium* is capable of producing short-chain fatty acids and some bioactive substances [28]. These bioactive compounds enhance the activity of digestive enzymes [29], stimulate the development of intestinal villi [30], and limit the proliferation of pathogenic bacteria [31]. By facilitating these processes, *Enterococcus faecium* has enhanced the nutritional digestibility of mink, thereby improving the growth performance of the animals.

In the current study, *Enterococcus faecium* increased the activity of SOD and effectively reduced the MDA content in the serum of minks. SOD serves as the first line of defense against oxygen-derived free radicals, catalyzing the conversion of superoxide to oxygen and hydrogen peroxide [32]. MDA is generated during the process of lipid peroxidation of unsaturated fatty acids within the phospholipids of the cell membrane [6]. A previous study has demonstrated the potential of *Enterococcus faecium* to increase the serum T-AOC level and SOD activity, as well as to decrease the serum MDA concentration in broilers [33]. In vitro studies have also found that *Enterococcus faecium* and its metabolites possess antioxidant activity [34,35,36]. Maintaining redox balance is essential for the health of cells and animals [37]. The antioxidant capacity of the organism represents its ability to resist oxidative damage [38]. *Enterococcus faecium* may enhance antioxidant function by inducing the secretion of bilirubin, which inhibits serum lipid peroxidation and the formation of reactive oxygen species (ROS) [39]. Furthermore, it may stimulate the expression of enzymes in the antioxidant defense system by activating and transferring nuclear factors, thereby promoting the removal of ROS and improving antioxidant capacity [40].

The immune response is an important defense mechanism in animals [41]. Immunoglobulins in serum are proteins produced by plasma cells and possess antibody activity [38]. They constitute the primary immune molecule of animal humoral immunity and are integral to the immune system [42]. The findings of the present study demonstrated that *Enterococcus faecium* increased IgA and IgG contents in the serum of growing male minks. Similarly, previous studies have found that *Enterococcus faecium* increases the concentration of IgG in the serum of broilers and increases the concentration of IgA in pigeon milk [10,43]. These results suggest that *Enterococcus faecium* has the potential to increase the concentration of various immunoglobulins (IgA, IgG, and IgM), thereby improving the humoral immune status of animals. It is possible that the heterologous antigens present in *Enterococcus faecium* stimulate the immune system, enhancing the production of immunoglobulins and improving humoral immunity [44]. 

In the present study, *Enterococcus faecium* also enhanced the level of SIgA in the jejunum mucosa, consistent with previous studies on broilers, which showed that dietary supplementation with *Enterococcus faecium* increased intestinal SIgA concentration during *Salmonella* and *Campylobacter* infection [45,46]. The mucosal system, with SIgA as its primary defense, is the first line of immune defense [47,48]. SIgA plays a crucial role in maintaining the integrity of the mucosal barrier and preventing pathogen invasion by promoting immune exclusion in the intestinal tract [49,50,51]. The observed increase in SIgA may be attributed to the induction of polymeric immunoglobulin receptor (pIgR) expression by *Enterococcus faecium*, which activates pattern recognition receptors and subsequently enhances SIgA secretion in the intestine [50]. 

Furthermore, *Enterococcus faecium* decreased IL-8 and IL-1β levels in the jejunal mucosa, as found in previous studies on porcine epithelial cells and broilers [52,53]. Both IL-1β and IL-8 are pro-inflammatory and play a key role in intestinal resistance to foreign pathogens [54,55]. Mi et al. [56] also reported in a study on broilers that *Enterococcus faecium* could reduce the production and secretion of IL-1β by preventing the expression of pro-Caspase-1. Cytokines are closely related to mucosal immune responses. Classified by their opposing effects on the inflammatory process, cytokines are divided into pro-inflammatory cytokines, which amplify the response, and anti-inflammatory cytokines, which mitigate it [47]. Disruption of the balance between pro-inflammatory and anti-inflammatory cytokines leads to the occurrence of inflammatory reactions [57]. *Enterococcus faecium*’s unique peptidoglycan composition, processed by peptidoglycan hydrolase secreting antigen A (SagA), produces muropeptides that trigger nucleotide-binding oligomerization domain-containing protein 2, thereby activating innate immunity and creating a micro-environment conducive to immunological therapy [58]. This mechanism may contribute to the immune-modulating effects of *Enterococcus faecium* on growing male minks.

Numerous studies have demonstrated that probiotics can exert beneficial effects by regulating the intestinal microbiota [59,60,61]. However, in this study, *Enterococcus faecium* had no effects on alpha diversity indices in minks. In contrast to this finding, a previous study in piglets reported that *Enterococcus faecium* increased various alpha diversity indices from day 1 to 14 [9]. Similarly, another study on broilers found that *Enterococcus faecium had* significant effects on the alpha diversity index at day 39 but not at day 21 [62]. These discrepancies may be due to the varying effects of *Enterococcus faecium* on different animal species and at different growth stages. We speculate that the increase in microbial species could lead to improved intestinal health [63]. The results of the Venn diagram in the current study have shown that *Enterococcus faecium* increased the species numbers at the OTU level, confirming its regulatory effect on the intestinal flora of growing male minks.

At the phylum level, *Proteobacteria* and *Firmicutes* were the most dominant on the rectal mucosa in male minks, which is consistent with previous studies on minks [64,65]. *Firmicutes*, known for their short-chain fatty acid (SCFA) production, can effectively regulate the intestinal immune system and maintain the balance of the intestinal flora [66]. In contrast, *Proteobacteria*, often associated with intestinal flora imbalance due to their production of lipopolysaccharide and flagellin, can lead to inflammatory responses [67]. 

At the genus level, *Enterococcus faecium* increased the relative abundance of *Paraclostridium*, *Brevinema*, and *Comamonas*, whose increase was also reported in a previous study on hybrid snakeheads [68]. *Comamonas*, an opportunistic pathogen found in various environments, including animal intestines [69], and *Brevinema*, a spirochaete involved in lignocellulose breakdown and nitrogen fixation [70]. In addition, *Paraclostridium* and *Brevinema* were the dominant bacteria in the intestinal flora of healthy fish [71,72]. These results suggest that *Enterococcus faecium* may promote metabolism and growth by regulating the relative abundance of intestinal flora. Moreover, our analysis of the correlation between the intestinal microbiota at the genus level and gut immune indicators revealed that *Enterococcus faecium* could act as gut microbiota modulators, enhancing the immune function of growing male minks and thereby improving the host’s defense against pathogenic microorganisms.

## 5. Conclusions

In conclusion, dietary supplementation with *Enterococcus faecium* (containing viable *Enterococcus faecium* at more than 10^7^ cfu/kg of diet) promotes growth performance, enhances the apparent digestibility of nutrients, and improves antioxidant capacity in growing male minks. Additionally, *Enterococcus faecium* also improves immunity and regulates the intestinal microbiota. However, this current study is a preliminary feeding experiment to investigate the probiotic effects of the isolated *Enterococcus faecium* on minks. Further research is necessary for an accurate supplementation evaluation and a comprehensive safety assessment of the isolated strain of *Enterococcus faecium*. This includes screening for virulence genes and antibiotic resistance to determine its potential as a probiotic candidate.

## Figures and Tables

**Figure 1 animals-14-02120-f001:**
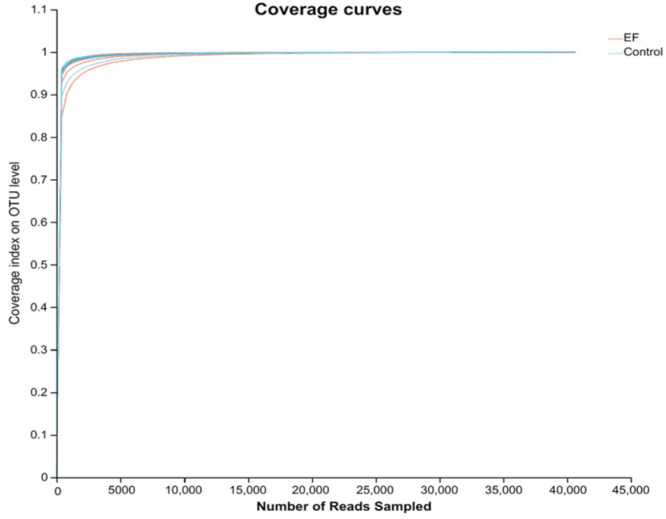
Coverage index of the two groups on the OUT level.

**Figure 2 animals-14-02120-f002:**
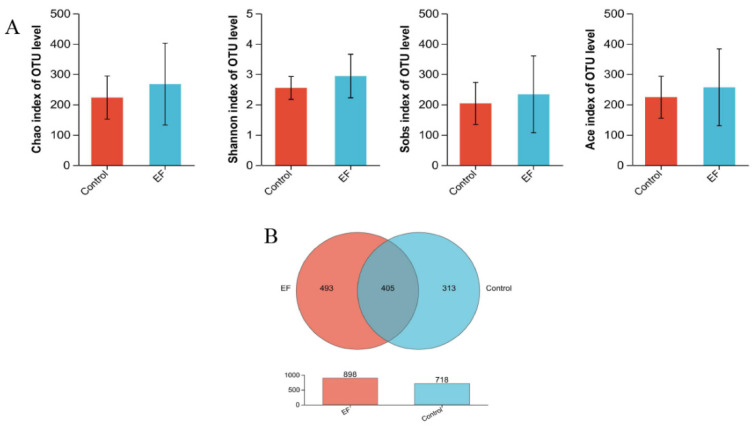
(**A**) Alpha diversity indices of the growing male minks in different groups, including the Chao index, Shannon index, Sobs index, and ACE index. (**B**) Venn diagram of species numbers in two groups at the OUT level.

**Figure 3 animals-14-02120-f003:**
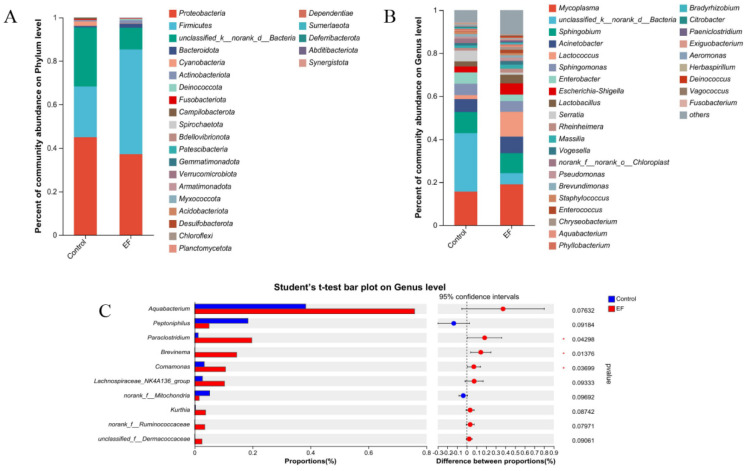
(**A**) Distribution of bacterial community structure at phylum level. (**B**) Distribution of bacterial community structure at the genus level. (**C**) The significance of differences among the two groups of the same species at the genus level (* represents *p* < 0.05). The result was statistically analyzed through the Student’s *t*−test.

**Figure 4 animals-14-02120-f004:**
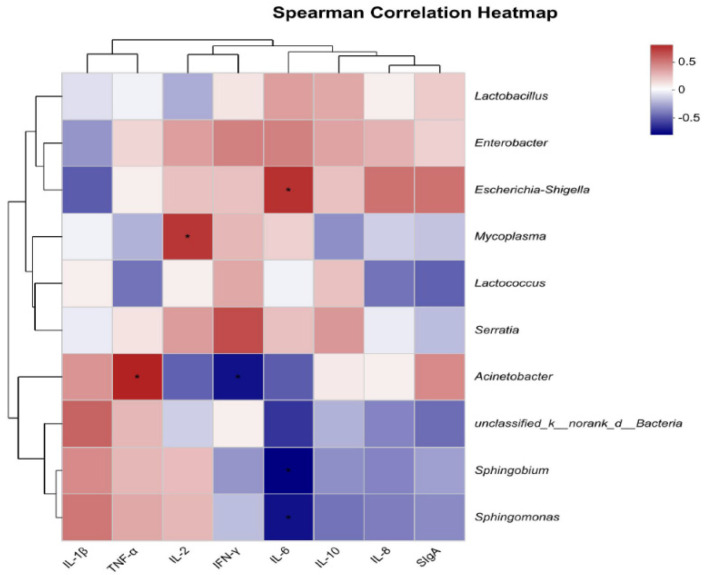
Heatmap shows the correlation between intestinal flora (genus level) and intestinal immune indicators. The X-axis and Y-axis are intestinal immune indicators and species, respectively, and the correlation R−values and *p*−values are obtained through calculation. R−values are displayed in different colors in the figure. If the *p*−values are less than 0.05, they are marked with *. The legend on the right is the color range of different R−values; the left and upper sides present the species and immune indicator cluster trees; * represents *p* < 0.05.

**Table 1 animals-14-02120-t001:** Composition and nutrient levels of diets (air-dry basis, %).

Items	0–4 Weeks	5–8 Weeks
Sea fish and byproducts	32	32
Unhatched fertilized egg	32	32
Chicken head	20	20
Extruded corn	10	10
Lard	1	2
Soybean meal	2	2
premix ^1^	3	2
Total	100	100
Nutrient levels		
ME (MJ/kg) ^2^	15.98	17.04
Ether extract	16.65	19.85
Crude protein	31.81	31.26
Calcium	2.47	2.59
phosphorus	1.59	1.64

^1^ The premix provided the following per kg of the diets: VA 9000 IU, VC 40 mg, VE 20 mg, VK_3_ 0.5 mg, VB_1_ 5 mg, VB_2_ 3 mg, VB_6_ 2.5 mg, VB_12_ 1 mg, VD_3_ 2000 IU, VB_3_ 20 mg, VB_5_ 6 mg, VB_9_ 0.5 mg, VB_7_ 0.5 mg, Fe 30 mg, Zn 25 mg, Mn 10 mg, Cu 5 mg, I 0.25 mg, Se 0.2 mg. ^2^ The metabolic energy values are derived from calculations, whereas the remaining values are obtained through direct measurement.

**Table 2 animals-14-02120-t002:** Effect of *Enterococcus faecium* on the growth performance of mink.

Items	Groups		*p*-Value
	0 (Control)	EF	
BW, g			
wk 0	1281.15 ± 45.60	1280.35 ± 40.03	0.962
wk 4	1736.54 ± 112.22	1825.38 ± 54.45	0.020
wk 8	2171.73 ± 167.59	2282.31 ± 84.09	0.044
ADG, g			
0–4 weeks	16.26 ± 3.66	19.46 ± 1.22	0.010
5–8 weeks	15.54 ± 3.79	16.32 ± 2.97	0.566
0–8 weeks	15.90 ± 2.86	17.89 ± 1.51	0.037
ADFI, g			
0–4 weeks	268.80 ± 12.67	267.95 ± 15.32	0.880
5–8 weeks	304.81 ± 21.54	295.15 ± 22.06	0.270
0–8 weeks	286.80 ± 14.12	281.55 ± 14.50	0.359
F/G			
0–4 weeks	17.36 ± 4.23	13.80 ± 1.05	0.011
5–8 weeks	20.73 ± 5.19	18.52 ± 2.75	0.188
0–8 weeks	18.60 ± 3.54	15.81 ± 1.10	0.016

BW = body weight, ADG = average daily gain, ADFI = average daily feed intake, and F/G = feed-to-gain ratio.

**Table 3 animals-14-02120-t003:** Effect of *Enterococcus faecium* on the apparent digestibility of nutrients in mink.

Items	Groups		*p*-Value
	0 (Control)	EF	
DM, %	75.59 ± 1.24	79.30 ± 0.60	<0.001
CP, %	86.40 ± 0.50	88.75 ± 0.59	<0.001
EE, %	96.78 ± 0.88	96.91 ± 1.56	0.867
Ash, %	19.26 ± 4.42	22.71 ± 5.76	0.320

DM = dry matter; CP = crude protein; EE = ether extract.

**Table 4 animals-14-02120-t004:** Effect of *Enterococcus faecium* on the antioxidant capacity of mink.

Items	Groups		*p*-Value
	0 (Control)	EF	
T-AOC, U/mL	17.28 ± 2.70	17.88 ± 3.24	0.713
MDA, nmol/mL	10.92 ± 1.41	9.24 ± 0.96	0.037
GSH-pX, μmol/L	1609.30 ± 219.50	1610.81 ± 221.78	0.991
SOD, U/mL	112.89 ± 24.63	144.18 ± 12.54	0.011

T-AOC = total antioxidant capacity; MDA = malondialdehyde; GSH-pX = glutathione peroxidase; SOD = superoxide dismutase.

**Table 5 animals-14-02120-t005:** Effect of *Enterococcus faecium* on serum immune indexes of mink.

Items	Groups		*p*-Value
	0 (Control)	EF	
IgA, μg/mL	57.38 ± 2.96	61.93 ± 1.90	0.008
IgG, g/L	6.58 ± 0.31	7.15 ± 0.31	0.005
IgM, μg/mL	504.51 ± 28.53	502.03 ± 51.31	0.907

IgA = immunoglobulin A; IgG = immunoglobulin G; IgM = immunoglobulin M.

**Table 6 animals-14-02120-t006:** Effect of *Enterococcus faecium* on the jejunum mucosal immune system of mink.

Items	Groups		*p*-Value
	0 (Control)	EF	
IL-2, p g/mL	340.43 ± 18.77	329.75 ± 39.14	0.592
IL-6, p g/mL	32.07 ± 4.74	35.72 ± 3.94	0.197
IL-8, p g/mL	116.58 ± 3.35	105.87 ± 2.79	0.001
IL-10, p g/mL	86.40 ± 9.73	84.82 ± 7.05	0.754
IL-1β, p g/mL	333.44 ± 27.23	299.47 ± 14.71	0.04
SIgA, p g/mL	2467.64 ± 100.81	3096.28 ± 154.19	<0.001
IFN-γ, p g/mL	1305.58 ± 295.82	1016.93 ± 77.63	0.094
TNF-α, p g/mL	764.83 ± 38.05	758.62 ± 86.23	0.875

IL-2 = interleukin-2; IL-6 = interleukin-6; IL-8 = interleukin-8; IL-10 = interleukin-10; IL-1β = interleukin-1β; SIgA = secretory immunoglobulin A; IFN-γ = interferon-gamma; TNF-α = tumor necrosis factor-alpha.

## Data Availability

The original contributions presented in the study are included in the article, further inquiries can be directed to the corresponding author.

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
