# Peer review of "Effects of Mink-Origin Enterococcus faecium on Growth Performance, Antioxidant Capacity, Immunity, and Intestinal Microbiota of Growing Male Minks"

_animals, 2024, doi:10.3390/ani14142120_

Round 1
Reviewer 1 Report (New Reviewer)
Comments and Suggestions for Authors
There are some minor typographical and grammatical errors such as:
1. Line no. 31 and elsewhere paraclostridium should be Paraclostridium
2. Line no. 119 instead of r/min should be g/min
3. Line no. 254 , (Comma) instead of . (Full stop)
4. Line no 274 prevented should be changed appropriately.
The results of the study showed that Enterococcus faecium could improve
a. the growth performance
b. increase the antioxidant capacity
c. Improve immunity of growing male minks,
d. Modulate the gut microbiota.
However results of this study were similar (a, b and c) to poultry and pigs but were dissimilar for gut microbiota change (d) which is surprising and suggesting that the effect produced by EF could also be replicated if we use similar firmicute antigen too.
All the parameters such as IgG and IgA increased in EF fed group which makes sense as we are feeding bacteria orally. However why IgM did not increased in some group which might be due to prior priming of the animals with EF. Is it so???
Comments on the Quality of English Language
The quality of English is good.
Author Response
Dear Reviewer,
Thank you for your comments concerning our manuscript entitled “Effects of Mink-origin Enterococcus faecium on Growth Performance, Antioxidant Capacity, Immunity and Intestinal Flora of Growing Male Minks” (ID: 3096722). Those comments are all valuable and very helpful for revising and improving our paper, as well as the important guiding significance to our researches. We have studied the comments carefully and have made correction which we hope meet with your approval. Revised portion are marked in red in the paper. The main corrections in the paper and the responds to comments are as following:
Responds to the reviewer's comments:
1 Line no. 31 and elsewhere paraclostridium should be Paraclostridium
Response: Corrections have been made as recommended. The revisions are highlighted in red on the revised paper (Line 32, 223, 225, 231, 339, 343, 345 and 349 in the revised version).
2 Line no. 119 instead of r/min should be g/min
Response: Corrections have been made as recommended. The revisions are highlighted in red on the revised paper (Line 140 and line 144 in the revised version).
3 Line no. 254 , (Comma) instead of . (Full stop)
Response: Corrections have been made (Line 296 in the revised version).
4 Line no 274 prevented should be changed appropriately.
Response: Corrections have been made (Line 317 in the revised version).
5 The results of the study showed that Enterococcus faecium could improve
- the growth performance
- increase the antioxidant capacity
- Improve immunity of growing male minks,
- Modulate the gut microbiota.
However results of this study were similar (a, b and c) to poultry and pigs but were dissimilar for gut microbiota change (d) which is surprising and suggesting that the effect produced by EF could also be replicated if we use similar firmicute antigen too.
All the parameters such as IgG and IgA increased in EF fed group which makes sense as we are feeding bacteria orally. However why IgM did not increased in some group which might be due to prior priming of the animals with EF. Is it so???
Response: It is a good idea. Further research is necessary to determine whether similar firmicute antigen exhibit similar effects to those of EF.
For the IgM, to my knowledge, IgM is a rapid responder with a short half-life, so it is possible that our experimental design did not capture the dynamic changes in IgM levels.
We tried our best to improve the manuscript and made some changes in the manuscript. These changes will not influence the content and framework of the paper. And here we did not list the changes but the revisions are highlighted in red on the revised paper.
We appreciate the editor and reviewer's earnest and warm work, and hope that the corrections will meet with approval. Please let me know if you have any questions.
Once again, thank you very much for your comments and suggestions.
Regards
Lin Cao
College of Animal Science and Technology
Qingdao Agricultural University
Shandong, China

Reviewer 2 Report (Previous Reviewer 1)
Comments and Suggestions for Authors
General comments:
The research paper entitled “Effects of Mink-origin Enterococcus faecium on Growth Performance, Antioxidant Capacity, Immunity and Intestinal Flora of Growing Male Minks” discussed the effects of supplementation of probiotic- Enterococcus faecium on performance of male mink. Although there are many research reports on use of Enterococcus faecium as probiotics in poultry and swine, but limited information is available on Mink. Therefore, the present study may be useful to find out the alternatives to AGP in mink.
Specific comments:
S1. Single dose of Enterococcus faecium was used in this study. How it was finalized?
22. Did authors follow any feeding standards for diet preparation? Please mention it.
T3. The experimental diet contained about 2.5% Ca and 1.5% P, which are quite high comparted to farm animals.
S4. Suggest to include data on digestibility of nutrients, which can substantiate the improved growth performance.
Author Response
Dear Reviewer,
Thank you for your comments concerning our manuscript entitled “Effects of Mink-origin Enterococcus faecium on Growth Performance, Antioxidant Capacity, Immunity and Intestinal Flora of Growing Male Minks” (ID: 3096722). Those comments are all valuable and very helpful for revising and improving our paper, as well as the important guiding significance to our researches. We have studied the comments carefully and have made correction which we hope meet with your approval. Revised portion are marked in red in the paper. The main corrections in the paper and the responds to comments are as following:
Responds to the reviewer's comments:
1 Single dose of Enterococcus faecium was used in this study. How it was finalized?
Response: We referred to our previous research on the effects of Lactobacillus plantarum on minks to establish the dosage of Enterococcus faecium in this experiment. The results of the previous studies have proven that supplementing the feed with the same concentration of Lactobacillus plantarum can have a positive impact on minks. Additionally, the results of this experiment support the chosen concentration and align with our research hypotheses.
2 Did authors follow any feeding standards for diet preparation? Please mention it.
Response: Information has been supplemented as recommended (Line 94-95 in the revised version)
3 The experimental diet contained about 2.5% Ca and 1.5% P, which are quite high compared to farm animals.
Response: We totally agree with the reviewer’s comments. However, the experiment was conducted in a commercial mink farm, and the diets were formulated on the farm. We are not permitted to change the formulation and feed ingredients. Although the Ca and P are quite high, the calcium-to-phosphorus ratio is reasonable. Furthermore, the content of Ca and P in diet should not impair the conclusion of this study because of the control.
4 Suggest to include data on digestibility of nutrients, which can substantiate the improved growth performance.
Response: According to the reviewer's suggestion, we have included data on the apparent digestibility of nutrients in revised manuscript. The revisions are highlighted in red on the revised paper (Including but not limited to the abstract, materials and methods, results, and discussion sections).
We tried our best to improve the manuscript and made some changes in the manuscript. These changes will not influence the content and framework of the paper. And here we did not list the changes but the revisions are highlighted in red on the revised paper. We appreciate the editor and reviewer's earnest and warm work, and hope that the corrections will meet with approval. Please let me know if you have any questions.
Once again, thank you very much for your comments and suggestions.
Regards
Lin Cao
College of Animal Science and Technology
Qingdao Agricultural University
Shandong, China

Reviewer 3 Report (New Reviewer)
Comments and Suggestions for Authors
General Comment
The Article describe the potential of Enterococcus faecium as a probiotic in Growing Male Minks. The research is focused on feeding trial by evaluating the growth performance, antioxidant capacity, immunity and intestinal flora of treated animals. The results are important, well presented. Discussion is really well written. However, there are several issues that need to be corrected. In addition, similar article has already been published in the Journal Frontiers in Veterinary Science (doi.org/10.3389/fvets.2024.1409127), so the authors should provide comment about difference in the data presented in this article.
After evaluating these Issues, I recommend it for publishing in your Journal.
Specific Comments.
Abstract.
Page 1, Line 31. Please check if the word “paraclostridium” should be capitalized!
Page 2, Lines 71-74. Is there any additional information about the approval? Some number and date of approval by mentioned Committee? This data should also be included here.
Page 2, Lines 75-80. Did the authors check the isolated strains for the presence of virulence genes? If so, how and which ones? Or is this information available from other sources? Considering this trial, this information is of high importance. Also, include full name of “MRS” in the text.
Page 2, Lines 81-89. It is not clear from the text how did the authors prepare the bacteria for use as “probiotic” in this trial. Please provide missing information.
Page 4, Lines 122-123. Please provide information on how the samples were homogenized.
Page 4, Lines 124-127. Please include the names of “kits” used for measurements.
Page 4, Line 134. Shouldn’t “Promega” be capitalized?
Page 4, Lines 131-135. It is not clear from the text whether these procedures were made by the research group or somewhere else. Please comment.
Page 4, Line 149. Please correct the typing error “week8”
Page 5, Line 160. Table 3. Please include information about the abbreviations used in the Table adding the text below the Table (as in previous one). The same is valid for other tables (Tables 4, 5)
Page 6, Line 185-193. Shouldn’t “proteobacteria” and “paraclostridium” be capitalized? Please check. Also, check the rest of the Article for this.
Page 8, Lines 212-313. Why is the text written here in different size?
Page 8, Lines 213-217. How is this finding related to the actual age of the animals? Please comment. It might also be useful to provide information of their age here.
Page 8, Lines 232-233. The sentence is a bit confusing. Please rewrite.
Page 8, Lines 248-249. Only against environmental stressors? Please comment.
Page 8-9, Lines 254-275. Please check grammar and style and correct typing errors. Also, be more specific when stating “similar effects”.
Page 9, Lines 290-293. Please check grammar and style and correct typing errors. Are there any similar studies in this animal species? Please comment.
Page 10, Lines 316-320. The Conclusion is too short. The sentence is a great summary, but please provide additional information that shows how this work was important and how this research support the use of this potential “pathogen” as probiotic.
Comments on the Quality of English LanguageThere are several issues that need to be corrected.
Author Response
Dear Reviewer,
Thank you for your comments concerning our manuscript entitled “Effects of Mink-origin Enterococcus faecium on Growth Performance, Antioxidant Capacity, Immunity and Intestinal Flora of Growing Male Minks” (ID: 3096722). Those comments are all valuable and very helpful for revising and improving our paper, as well as the important guiding significance to our researches. We have studied the comments carefully and have made correction which we hope meet with your approval. Revised portion are marked in red in the paper. The main corrections in the paper and the responds to comments are as following:
Responds to the reviewer's comments:
1 In addition, similar article has already been published in the Journal Frontiers in Veterinary Science (doi.org/10.3389/fvets.2024.1409127), so the authors should provide comment about difference in the data presented in this article.
Response: Thank you for the reviewer's attention to our research. We conducted two experiments simultaneously to explore the effects of live Enterococcus faecium and Enterococcus faecium-derived postbiotics on growing male minks, respectively. The differences in the data presented in the two articles are attributed to the variations among different individuals.
2 Page 1, Line 31. Please check if the word “paraclostridium” should be capitalized!
Response: Corrections have been made as recommended. The revisions are highlighted in red on the revised paper (Line 32 in the revised version).
3 Page 2, Lines 71-74. Is there any additional information about the approval? Some number and date of approval by mentioned Committee? This data should also be included here.
Response: The information about the approval has been added to the article. The revisions are highlighted in red on the revised paper (Line 76-77 in the revised version).
4 Page 2, Lines 75-80. Did the authors check the isolated strains for the presence of virulence genes? If so, how and which ones? Or is this information available from other sources? Considering this trial, this information is of high importance. Also, include full name of “MRS” in the text.
Response: At present, we have not conducted virulence gene detection on the isolated Enterococcus faecium. This study is a preliminary feeding experiment. Further research is necessary for a comprehensive risk assessment and safety evaluation of the Enterococcus faecium strain as a probiotic candidate. This includes screening for virulence genes. We have included the full name of MRS in the text (Line 82 in the revised version).
5 Page 2, Lines 81-89. It is not clear from the text how did the authors prepare the bacteria for use as “probiotic” in this trial. Please provide missing information.
Response: Information has been supplemented as recommended (Line 82-86 in the revised version).
6 Page 4, Lines 122-123. Please provide information on how the samples were homogenized.
Response: Information has been supplemented as recommended (Line 142-144 in the revised version).
7 Page 4, Lines 124-127. Please include the names of “kits” used for measurements.
Response: Corrections have been made as recommended. The revisions are highlighted in red on the revised paper (Line 147-153 in the revised version).
8 Page 4, Line 134. Shouldn’t “Promega” be capitalized?
Response: Corrections have been made as recommended. The revisions are highlighted in red on the revised paper (Line 160 in the revised version).
9 Page 4, Lines 131-135. It is not clear from the text whether these procedures were made by the research group or somewhere else. Please comment.
Response: The detection of the genetic sequences of gut microbiota was entrusted to Shanghai Major Bio-Pharmaceutical Technology Co., Ltd.
10 Page 4, Line 149. Please correct the typing error “week8”
Response: Corrections have been made as recommended. The revisions are highlighted in red on the revised paper (Line 175 in the revised version).
11 Page 5, Line 160. Table 3. Please include information about the abbreviations used in the Table adding the text below the Table (as in previous one). The same is valid for other tables (Tables 4, 5)
Response: Corrections have been made as recommended. The revisions are highlighted in red on the revised paper (Line 193-194, 200, 207-209 in the revised version).
12 Page 6, Line 185-193. Shouldn’t “proteobacteria” and “paraclostridium” be capitalized? Please check. Also, check the rest of the Article for this.
Response: Corrections have been made as recommended. The revisions are highlighted in red on the revised paper (Line 32, 223, 225, 231, 339, 343, 345 and 349 in the revised version).
13 Page 8, Lines 212-313. Why is the text written here in different size?
Response: Corrections have been made as recommended. The revisions are highlighted in red on the revised paper (Line 391-302 in the revised version).
14 Page 8, Lines 213-217. How is this finding related to the actual age of the animals? Please comment. It might also be useful to provide information of their age here.
Response: Yes, the age information of the experimental minks should indeed be useful. We have provided (Line 252 in the revised version).
15 Page 8, Lines 232-233. The sentence is a bit confusing. Please rewrite
Response: Corrections have been made as recommended. The revisions are highlighted in red on the revised paper (Line 276-277 in the revised version).
16 Page 8, Lines 248-249. Only against environmental stressors? Please comment.
Response: It's not merely a defense against environmental stressors. Corrections have been made as recommended. The revisions are highlighted in red on the revised paper (Line 291 in the revised version).
17 Page 8-9, Lines 254-275. Please check grammar and style and correct typing errors. Also, be more specific when stating “similar effects”.
Response: Corrections have been made as recommended. The revisions are highlighted in red on the revised paper (Line 396-318 in the revised version).
18 Page 9, Lines 290-293. Please check grammar and style and correct typing errors. Are there any similar studies in this animal species? Please comment.
Response: We have reviewed the grammar and style and corrected typing errors. The revisions are highlighted in red on the revised paper (Line 336 in the revised version). There are studies related to the impact of other additives (Lactobacillus plantarum, antimicrobial peptides, glucose oxidase, poly-lysine, etc) on the gut microbiota of minks.
19 Page 10, Lines 316-320. The Conclusion is too short. The sentence is a great summary, but please provide additional information that shows how this work was important and how this research support the use of this potential “pathogen” as probiotic.
Response: Corrections have been made as recommended. The revisions are highlighted in red on the revised paper (Line 362-366 in the revised version).
We tried our best to improve the manuscript and made some changes in the manuscript. These changes will not influence the content and framework of the paper. And here we did not list the changes but the revisions are highlighted in red on the revised paper. We appreciate the editor and reviewer's earnest and warm work, and hope that the corrections will meet with approval. Please let me know if you have any questions.
Once again, thank you very much for your comments and suggestions.
Regards
Lin Cao
College of Animal Science and Technology
Qingdao Agricultural University
Shandong, China

Reviewer 4 Report (New Reviewer)
Comments and Suggestions for Authors
Comments
1. Is it accurate to classify enterococcus as lactic acid bacteria in the article? Do the probiotics produced by enterococcus meet the standards of lactic acid bacteria?
2. The experimental strain was collected from the rectal contents of mink, why not consider enterococcus from other intestinal segments? Is it different for different segments of the intestine?
3. Are the collected strains of mink and the experimental mink the same species? Is it male or female, what age, do those factors affect the experiment?
4. Do you select one mink or multiple minks for the collection of strains? What is the basis for the selection?
5. How is the amount of Enterococcus faecium added to the mink's feed in the experimental operation determined? Is there any basis?
6. Enterococcus is also a conditioned pathogen. By increasing the amount of enterococcus in the form of feed addition to mink, will it have other negative effects on mink in some special circumstances?
7. In the discussion part, the 12-week-old mink was selected for the experiment. Does this age belong to the early growth and development of minks?
8. In the discussion section, it was described that Enterococcus faecium had no effect on the α diversity index of mink. However, it can be observed from Figure 2 that all indices of α diversity in the experimental group were higher than those in the control group, indicating a certain influence, why?
9. In the discussion section, it is mentioned that Enterococcus faecium has an effect on α diversity in piglets and broilers. Are there any references for this conclusion? It's not marked.
10. Conclusion: Adding live bacteria Enterococcus faecium to concentrations greater than 10-7 cfu/kg is beneficial to minks, but what are the effects of higher or lower concentrations of bacteria?
Author Response
Dear Reviewer,
Thank you for your comments concerning our manuscript entitled “Effects of Mink-origin Enterococcus faecium on Growth Performance, Antioxidant Capacity, Immunity and Intestinal Flora of Growing Male Minks” (ID: 3096722). Those comments are all valuable and very helpful for revising and improving our paper, as well as the important guiding significance to our researches. We have studied the comments carefully and have made correction which we hope meet with your approval. Revised portion are marked in red in the paper. The main corrections in the paper and the responds to comments are as following:
Responds to the reviewer's comments:
1 Is it accurate to classify enterococcus as lactic acid bacteria in the article? Do the probiotics produced by enterococcus meet the standards of lactic acid bacteria?
Response: Thank you very much for the reviewer's question. Many articles have indicated that Enterococcus can be classified as lactic acid bacteria (Franz et.al., 2011, Nascimento et.al., 2019). In particular, Enterococcus faecium was approved by the Ministry of Agriculture of China for use as a direct-fed microbial in 2003. Additionally, numerous animal experiments have confirmed that Enterococcus faecium possesses the general characteristics and health benefits associated with lactic acid bacteria probiotics (Hu et.al., 2019, Wu et.al., 2019).
Franz M C ,Huch M ,Abriouel H , et al.Enterococci as probiotics and their implications in food safety [J].International Journal of Food Microbiology,2011,151(2):125-140.
Nascimento S C L ,Casarotti N S ,Todorov D S , et al.Probiotic potential and safety of enterococci strains[J].Annals of Microbiology,2019,69(3):241-252.
Hu, C.; Xing, W.; Liu, X.; Zhang, X.; Li, K.; Liu, J.; Deng, B.; Deng, J.; Li, Y.; Tan, C. Effects of dietary supplementation of probiotic Enterococcus faecium on growth performance and gut microbiota in weaned piglets. Amb Express. 2019, 9, 1-12.
Wu, Y.; Zhen, W.; Geng, Y.; Wang, Z.; Guo, Y. Effects of dietary Enterococcus faecium NCIMB 11181 supplementation on growth performance and cellular and humoral immune responses in broiler chickens. Poultry Sci. 2019, 98, 150-163.
2 The experimental strain was collected from the rectal contents of mink, why not consider enterococcus from other intestinal segments? Is it different for different segments of the intestine?
Response: In fact, it is not obvious to identify the intestinal segments of the mink, except for the duodenum and cecum. Furthermore, the rectal content is relatively easy to sample and contains a high abundance and diversity of microorganisms. The microbial community in the rectal content may have a higher degree of stability both between individuals and within the same individual. Therefore, we chose the rectal content as the experimental material.
The composition and relative abundance of microorganisms in different intestinal segments may differ significantly due to the varying environmental conditions in each segment. Further research is still required to identify these detailed results.
3 Are the collected strains of mink and the experimental mink the same species? Is it male or female, what age, do those factors affect the experiment?
Response: Yes, the collected strains of mink and the experimental mink belong to the same species. The Enterococcus faecium strains were isolated from the rectal contents of healthy, growing male minks. It is probable that breed, sex, and age influence the experiment. However, this should not impair the conclusions of this study due to the experimental design.
4 Do you select one mink or multiple minks for the collection of strains? What is the basis for the selection?
Response: The rectal contents Collected from several minks and cultured them using MRS medium (a specific medium for lactic acid bacteria), respectively. These suspected lactic acid bacterial strains were purified and identified through 16S rRNA analysis to confirm the strains. Previous experiments have shown a positive correlation between lactic acid bacteria in the mink intestine and the animal's immune system. This has led to the scientific question of whether the exogenous supplementation of mink-origin lactic acid bacteria could be beneficial to the health of minks. Therefore, this experiment was designed to address this question. The experimental results demonstrated that the mink-origin Enterococcus faecium was beneficial to minks.
5 How is the amount of Enterococcus faecium added to the mink's feed in the experimental operation determined? Is there any basis?
Response: We referred to our previous research on the effects of Lactobacillus plantarum on minks to establish the dosage of Enterococcus faecium in this experiment. The results of the previous studies have proven that supplementing the feed with the same concentration of Lactobacillus plantarum can have a positive impact on minks. Additionally, the results of this experiment support the chosen concentration and align with our research hypotheses.
6 Enterococcus is also a conditioned pathogen. By increasing the amount of Enterococcus in the form of feed addition to mink, will it have other negative effects on mink in some special circumstances?
Response: Enterococcus faecium can become opportunistic pathogens under certain conditions, but Amaral et al. have suggested that Enterococcus faecium strains isolated from food, commensal organisms, and the environment pose a low risk. Consequently, we selected Enterococcus faecium strains isolated from healthy minks as our experimental material. Moreover, our experimental results indicate no adverse health effects, suggesting that the addition of Enterococcus faecium from the healthy mink gut is safe under the current experimental conditions.
Amaral, D.M.F.; Silva, L.F.; Casarotti, S.N.; Nascimento, L.C.S.; Penna, A.L.B. Enterococcus faecium and Enterococcus durans isolated from cheese: Survival in the presence of medications under simulated gastrointestinal conditions and adhesion properties. J Dairy Sci. 2017, 100, 933-949.
7 In the discussion part, the 12-week-old mink was selected for the experiment. Does this age belong to the early growth and development of minks?
Response: According to 'Nutrition and Nutritional Physiology of the Mink' written by Leoschke W. L., 12-week-old minks are considered to be in the late growth stage.
8 In the discussion section, it was described that Enterococcus faecium had no effect on the α diversity index of mink. However, it can be observed from Figure 2 that all indices of α diversity in the experimental group were higher than those in the control group, indicating a certain influence, why?
Response: Yes, it can be observed from Figure 2 that all indices of α-diversity in the experimental group are higher than those in the control group. In statistics, The principle of statistics is based on the principle that low-probability events do not occur. A P-value greater than 0.05 suggests that the observed difference may not necessarily be due to EF, but could be the result of error, and therefore it cannot be concluded that EF causes an increase in the α-diversity of the mink gut microbiota.
9 In the discussion section, it is mentioned that Enterococcus faecium has an effect on α diversity in piglets and broilers. Are there any references for this conclusion? It's not marked.
Response: The discussion section mentions the impact of Enterococcus faecium on the α-diversity of piglets and broilers. There are references to support this conclusion, specifically References 9 (line 332) and Reference 62 (line 333).
10 Conclusion: Adding live bacteria Enterococcus faecium to concentrations greater than 107cfu/kg is beneficial to minks, but what are the effects of higher or lower concentrations of bacteria?
Response: We have refined the discussion section of the paper. The revisions are highlighted in red on the revised paper (Lines 362-366).
We tried our best to improve the manuscript and made some changes in the manuscript. These changes will not influence the content and framework of the paper. And here we did not list the changes but the revisions are highlighted in red on the revised paper. We appreciate the editor and reviewer's earnest and warm work, and hope that the corrections will meet with approval. Please let me know if you have any questions.
Once again, thank you very much for your comments and suggestions.
Regards
Lin Cao
College of Animal Science and Technology
Qingdao Agricultural University
Shandong, China

Round 2
Reviewer 4 Report (New Reviewer)
Comments and Suggestions for Authors
No comments.
This manuscript is a resubmission of an earlier submission. The following is a list of the peer review reports and author responses from that submission.
Round 1
Reviewer 1 Report
Comments and Suggestions for Authors
L72: Add- How postbiotic was applied in diet” Is it in semisolid or dry form?
L104: Add reference for ME calculation.
L114: ” Nutrients Apparent Digestibility” change to “ Determination of apparent nutrient digestibility”
L115: “valuate” -??
L116-119: “A total of 24 un-contaminated feces samples in the 4 groups (with 6 replicates in each group) were sampled approximate 200 grams in three days, respectively. The 3-day feces samples were mixed, then kept at −20 ℃ until analysis. Meanwhile, the diets of each group were sampled daily during the 3 days before they were fed to the minks,”----- Rewrite the sentences and add reference for “endogenous indicator method”.
L 184-185: “Enterococcus faecium postbiotics had significant effects on the apparent digestibility of CP , EE and DM (all P<0.05, Table 3).”-However, digestibility of DM on 4 wks was not affected. Revise it.
Discussion section: Explanation is required for why responses were better in 0.1% PEF group compared to 0.05 and 0.15% PEF groups.